# The Impact of Drug Interactions in Patients with Community-Acquired Pneumonia on Hospital Length of Stay

**DOI:** 10.3390/geriatrics7010011

**Published:** 2022-01-04

**Authors:** Johannes Peter Schmitt, Andrea Kirfel, Marie-Therese Schmitz, Hendrik Kohlhof, Tobias Weisbarth, Maria Wittmann

**Affiliations:** 1Department of Anaesthesiology and Intensive Care Medicine, University Hospital Bonn, Venusberg-Campus 1, 53127 Bonn, Germany; johannespeterschmitt@gmail.com (J.P.S.); maria.wittmann@ukbonn.de (M.W.); 2Department of Medical Biometry, Informatics and Epidemiology, Medical Faculty, University of Bonn, Venusberg-Campus 1, 53127 Bonn, Germany; m.schmitz@imbie.meb.uni-bonn.de; 3Clinic and Polyclinic for Orthopaedics and Trauma Surgery, University Hospital Bonn, Venusberg-Campus 1, 53127 Bonn, Germany; UHO@antonius-koeln.de; 4Clinic for Cardiology and Rhythmology, St. Vinzenz Hospital Cologne, Merheimer Str. 221-223, 50733 Cologne, Germany; tweisbarth@me.com

**Keywords:** elderly patients, polypharmacy, drug interaction, multimorbidity, length of stay, community-acquired pneumonia

## Abstract

(1) Background: An aging society is frequently affected by multimorbidity and polypharmacy, which, in turn, leads to an increased risk for drug interaction. The aim of this study was to evaluate the influence of drug interactions on the length of stay (LOS) in hospitals. (2) Methods: This retrospective, single-centre study is based on patients treated for community-acquired pneumonia in the hospital. Negative binomial regression was used to analyse the association between drug interactions and the LOS in the hospital. (3) Results: The total cohort contained 503 patients, yet 46 inpatients (9%) that died were not included in the analyses. The mean age was 74 (±15.3) years, 35% of patients older than 65 years were found to have more than two drug interactions, and 55% had a moderate, severe, or contraindicated adverse drug reaction. The regression model revealed a significant association between the number of drug interactions (rate ratio (RR) 1.02; 95%-CI 1.01–1.04) and the severity of drug interactions (RR 1.22; 95%-CI 1.09–1.37) on the LOS for the overall cohort as well as for the subgroup of patients aged 80 years and older. (4) Conclusion: Drug interactions are an independent risk factor for prolonged hospitalisation. Standardised assessment tools to avoid drug interactions should be implemented in clinical routines.

## 1. Introduction

The world’s population aged 60 or older numbered 962 million in 2017, and the number is expected to double again to almost 2.1 billion by 2050 [1]. Thus, it is clear that the diseases of the elderly will increasingly determine the health system of the future [2]. Multimorbidity and its consequences will lead to a global health problem in the foreseeable future [3,4,5]. The Federal Ministry of Education and Research already published in 2014 that almost two-thirds (62%) of people over 65 suffer from at least three chronic diseases and thus fulfil the criterion of multimorbidity [6]. In Great Britain, even 70% of inpatient hospital admissions were attributed to long-term health conditions [7]. Due to mounting multimorbidity, the number of drug prescriptions is consequently going to rise as well. As it has already been shown in the literature, more than one-third of people over 65 take five or more medications at the same time with an upward trend [8]. In a clinical study on a geriatric patient cohort with multiple chronic diseases, in Saudi Arabia, it was shown that even 55% of patients were affected by polypharmacy [9]. With increasing polypharmacy again, the risk for the occurrence of drug interactions increases inevitably and could thus lead to multiple unwanted side effects for the patient. A review in 2020 indicated that multimorbidity and polypharmacy are associated with a progressive loss of resilience and impaired homeostasis contributing to a significant burden on health and social care [10]. Furthermore, a retrospective, single-centre study in 2017 revealed that patients with potential inappropriate discharge medications showed an increased risk for readmission [11]. In addition, there is evidence in the literature that serious adverse drug reactions are preventable [12]. 

Thus far, less attention has been shown to the extent to which the risk factor of drug interaction influences relevant outcomes, such as length of hospital stay. One study describes exactly the opposite effect, indicating that length of hospital stay is an independent predictor of adverse drug reaction [13]. However, length of stay is an issue that occurs to be of great importance regarding steadily rising treatment costs and ubiquitous nursing shortage [14,15,16]. A prospective study from the UK in 2005 with 3695 patients showed that almost 15% of patients experienced at least one adverse drug reaction during their stay in the hospital. Moreover, it could be shown that the average length of stay in the hospital was significantly longer in patients with at least one drug interaction [17].

Individuals who are older than 65 years or have a chronic health condition are considered to be at high risk for pneumonia. Pneumonia is the most common infectious disease of the industrial nations affecting geriatric, and multimorbid patients with multiple medications to a special degree [18,19]. Since patients with community-acquired pneumonia have risk factors (age and comorbidities) that are also considered risk factors for drug interactions, this cohort deserves special attention.

To assess the impact of drug interactions on relevant outcomes such as length of hospital stay, this retrospective study analyses adverse drug events that occurred in patients during treatment for community-acquired pneumonia.

In contrast to previous studies, this analysis considered the extent and influence of patients’ comorbidities to evaluate whether patients stay longer in the hospital mostly because of their multimorbidity, or whether adverse drug reactions have an independent impact on the length of stay in the hospital, especially in the elderly population.

## 2. Materials and Methods

### 2.1. Design

This is a retrospective, single-centre study based on patients with community-acquired pneumonia. The design is a cross-sectional study.

### 2.2. Sample and Setting

Data collection was conducted from inpatients in an internal medicine department of a hospital providing basic and regulated care from 2013 to 2018. The patient data analysed pertain only to the inpatient period and its discharge type. All patients hospitalised with community-acquired pneumonia during the specified period of time were included in this retrospective data analysis. Inclusion criteria were community-acquired pneumonia without necessary ventilation or intensive care unit stay. For a uniform and comparable treatment regimen, the criteria of the quality assurance procedure of the Institute for Quality Assurance and Transparency in Health Care (IQTIG) were applied [20]. The only exclusion criterion was having an age of below 18 years. Patients who died during the inpatient stay were removed from the analysis based on the lack of comparability of the outcome (length of stay). Based on the retrospective nature a of the study and anonymised data processing, no patient consent forms were available.

### 2.3. Data Collection

For each enrolled patient, six variables were collected. These included age and gender. In addition, the length of stay (LOS) in the hospital and the severity of inpatient treatment were evaluated for each patient in the form of the German Patient Clinical Complexity Level (PCCL). In the German Diagnosis Related Group (G-DRG) classification, complications and/or comorbidities (CCs) are mapped using the patient-related total severity code (PCCL). The PCCL is calculated from the cumulative severities of complications and/or comorbidities (CCL) of a patient’s individual [21]. It indicates the overall patient-related severity of the respective treatment case based on a value between 0 (no ‘patient clinical complexity’) and 6 (most severe ‘patient clinical complexity’). The PCCL score was applied as a surrogate parameter of comorbidities in this design. In addition, the number and severity of drug interactions were collected, which are described in more detail in the following section.

All medications received by the patient during their inpatient treatment were checked for potential pharmacokinetic and pharmacodynamic interactions using the AiD Klinik^®^ program of Dosing GmbH Heidelberg [22,23]. The basis of the interaction check performed by the AiD-Klinik^®^ program is an evidence-based database, which indicates interactions as soon as they are of clinical relevance and if they are supported by sufficient clinical data. Published interactions from case series and studies are stored in this database. New drug interaction data were searched on a weekly based PubMed screening of relevant journals. Professional information and red hand letters served as additional sources. Interactions without evidence or without clinically valid data were not entered. The number and severity of interactions that could be detected per patient during treatment were classified by the program according to the following colour scheme after an interaction check: ‘black’, contraindicated combination; ‘red’, an interaction that can cause serious adverse drug reactions; ‘orange’, an interaction that can cause moderate adverse drug reactions; ‘yellow’, a mild interaction that can statistically lead to significant changes but usually has no clinical relevance.

### 2.4. Data Analysis

Statistical analysis was carried out using the statistical programming environment R [24]. Descriptive statistics are shown as mean and standard deviation for quantitative variables. Qualitative variables are summarised by absolute and relative frequencies. Negative binomial regression was used to analyse the association of LOS and the number and severity of drug interactions, adjusted for age and PCCL [25]. The severity of drug interactions was dichotomised into ‘none/mild’ and ‘moderate/severe and/or contraindicated’. Rate ratios (RRs) with 95% confidence intervals were reported. Regression analysis was performed for the total cohort and separately for three different age groups (<65 years, 65–79 years, and 80 years and older).

### 2.5. Ethical Considerations

This retrospective, chart-review study involving human participants was in accordance with the ethical standards of the institutional and national research committee and with the 1964 Helsinki Declaration and its later amendments or comparable ethical standards. According to the professional Code of Conduct of the North Rhine Medical Association, ethical approval is not necessary for a retrospective analysis of routine data.

## 3. Results

This retrospective data analysis included 503 patients. Of those, 457 (91%) were discharged, and 46 (9%) patients died during their inpatient stay. Based on the lack of comparability between the discharge cohort and the dead patients, the deceased patients were removed from the dataset to be analysed. Characteristics of the included patients for the entire cohort and their age subdivisions are shown in Table 1. The gender distribution was similar with 224 (49%) men and 233 (51%) women. The average age of the entire cohort was 74 (±15.3) years. The average LOS of patients was 8.5 (±5.0) days. The majority (*n* = 136, 30%) of included patients showed a PCCL of 3. A similar number of patients (*n* = 134, 29%) were classified as PCCL category 0. The number of patients with a PCCL of 0 was most common in patients younger than 65 years (48%). The number of drug interactions differed between the age groups. Patients aged 65–79 years or 80 years and older were substantially more likely to have more than two interactions (36% and 33%, respectively) than patients of the youngest age group (11%). Furthermore, the two older subgroups showed higher prevalence rates of moderate, severe, or contraindicated drug interactions (50% and 60%) than younger patients (30%).

### 3.1. Impact of Drug Interaction on LOS for the Total Cohort

The aim of this study was to review various factors influencing the length of stay in a hospital and to determine whether drug interactions have a demonstrable effect on it. The average length of stay in the hospital increased in conjunction with the age of the patients. LOS was on average 7 days in patients younger than 65 years and on average 9 days in the two older age groups (≥65 years). Figure 1 shows the LOS by age using bar charts.

Table 2 shows the results of the regression analysis for the total cohort. LOS in the hospital was significantly related to all four independent variables. In comparison with patients with no or mild drug interactions, LOS increased by a factor of 1.22 (95%-CI: 1.09–1.37) for patients with at least one moderate-to-severe or contraindicated drug interaction. Further, LOS increased by 2% in the presence of one additional drug interaction (1.02; 95%-CI: 1.01–1.04).

### 3.2. Impact of Drug Interaction on LOS for the Cohort of ≥80 Years old

In addition to considering the influence of drug interaction for the total cohort, it was also analysed for the different age groups. There was no significant association between LOS and the number of drug interactions for patients aged <65 years or patients aged between 65 and 79 years (Appendix A). Equally, there was no significant effect on LOS for the severity of drug interaction in those aged 65 to 79 years. For patients aged less than 65 years, a significant association between LOS and drug interaction severity was observed (*p* = 0.003).

The results of the regression analysis for patients aged 80 years and older are presented in Table 3. The PCCL and the number of drug interactions showed a significant effect on LOS. For patients aged ≥80, LOS was prolonged by 3% per additional drug interaction (1.03; 95%-CI: 1.00–1.06). Furthermore, 31% of patients 80 years or older had at least one to two drug interactions, and 33% had more than two interactions. It can be concluded that among the elderly, 33% had a 9% increase in LOS due to at least three drug interactions. Of the patients who were 80 years of age or older, 60% had at least moderate-to-severe or even contraindicated drug interactions. For those, LOS increased by a factor of 1.18 (95%-CI: 0.99–1.40) in comparison with patients with no or mild drug interactions.

## 4. Discussion

This retrospective study shows an independent effect of drug interaction on hospital length of stay. During their inpatient hospitalisation, 35% of patients with an age older than 65 years had more than two drug interactions. This statement applies to the younger cohort for only 11% of those under 65 years. Furthermore, moderate, severe, or contraindicated drug interaction was evident in 55% of this cohort and only in 30% of the young cohort in return. Based on these findings, it can be concluded that the number and severity of drug interactions are not uncommon in hospitals and show an increasing trend with age. These findings have already been confirmed by the literature, and it has also been noted that the problem of drug interactions has continued to increase over the last several decades [26]. The higher number of secondary diagnoses explains the fact that the older generation is more frequently affected by drug interactions. This is also confirmed by the existing literature [27].

The results of the regression showed that both the number and severity of drug interactions are significantly associated with LOS after risk adjustment (of age and PCCL). LOS for the total patient cohort increased by a factor of 1.22 (95% CI: 1.09–1.37; *p* < 0.001) if the patient had at least one moderate-to-severe drug interaction. In addition, LOS increased by 2% (1.02; CI: 0.97–1.24; *p* = 0.005) in the presence of one drug interaction. Thus, it can be stated that patients’ health condition determines decisively the length of hospital stay, as one might suspect, but the extent of adverse drug reactions affects the LOS independently as well. This result has received little attention in the literature so far, and the results vary between 0.25 and 7 days [28,29]. However, it should be noted that the published sources have different patient cohorts (internal medicine and surgery). However, these studies, as well as our results, suggest that drug interactions have an impact on patients’ length of hospital stay.

Based on regression model data from the patients who were 80 years of age or older, LOS was significantly prolonged by 3% with at least one drug interaction (1.03; 95%-CI: 1.00–1.06). This is 1% more than for the total cohort. Furthermore, results from the same cohort show that the severity of drug interactions has an impact on LOS, with a factor of 1.18 for at least one moderate interaction (95%-CI: 0.99–1.40). These findings are also supported by the existing literature [28,29].

In summary, drug interactions represent an independent risk factor for prolonged hospitalisation and are thus a determining factor for economic efficiency in the health care system. In order to reliably prevent adverse drug reactions, digital drug programs such as the AID Klinik^®^, Heidelberg, Germany, must be used as the standard—a procedure that has not yet become widely established in daily routine. The intentions of Holt et al. from 2010 to avoid adverse drug interactions in geriatric patients by establishing the ‘Priscus list’ appear to be of continuing and current importance in view of the corresponding results of our study [30].

In view of the upcoming central challenges of future medicine—multimorbidity and polypharmacy, on the one hand, and rising treatment costs and the immanent nursing shortage, on the other—the central results of our data call for a more consistent consideration of adverse drug reactions as a standard in clinical and outpatient treatment. In view of the prospective efforts to avoid the proven consequences of drug interactions, the quality of drug interactions should be analysed in a more differentiated way in the future.

In this retrospective study, no distinction was made between drug interactions that were intentionally prescribed, such as acetylsalicylic acid and rivaroxaban, in the treatment of coronary heart disease, and avoidable drug interactions, such as atorvastatin and clarithromycin. In this regard, subsequent studies should differentiate between avoidable and unavoidable drug interactions. It is assumed that avoidable, relevant drug interactions could even have a greater impact on relevant outcomes, such as LOS, and would therefore highlight the importance of avoiding adverse drug reactions considering the issues of geriatric medicine and health economics in the future.

### Limitations

This retrospective data analysis has limitations that need to be mentioned here. On the one hand, the number of drugs taken per patient and a list of interactions are missing from the present raw dataset. These could not be collected in detail later. However, when using the AiD tool to generate the number and severity of drug interactions, these were completely available in the clinic. Furthermore, the known variables of multimorbidity using the PCCL score and age were included in the statistical analyses, but there is the possibility that other confounders not recorded here influence the outcome of length of stay. Although the AiD tool has been used in several clinical trials, a validation study for the tool itself has not been published.

## 5. Conclusions

Our results suggest that drug interactions in the cohort of patients with community-acquired pneumonia increase with age. This concerns the number of drug interactions during the inpatient stay, but also the intensity of the interaction. Furthermore, the occurrence of drug interactions is an independent factor influencing the length of hospital stay. These results suggest that standardised drug information systems should be used in routine clinical practice to minimise, or at best avoid, the complications that arise from drug interactions.

## Figures and Tables

**Figure 1 geriatrics-07-00011-f001:**
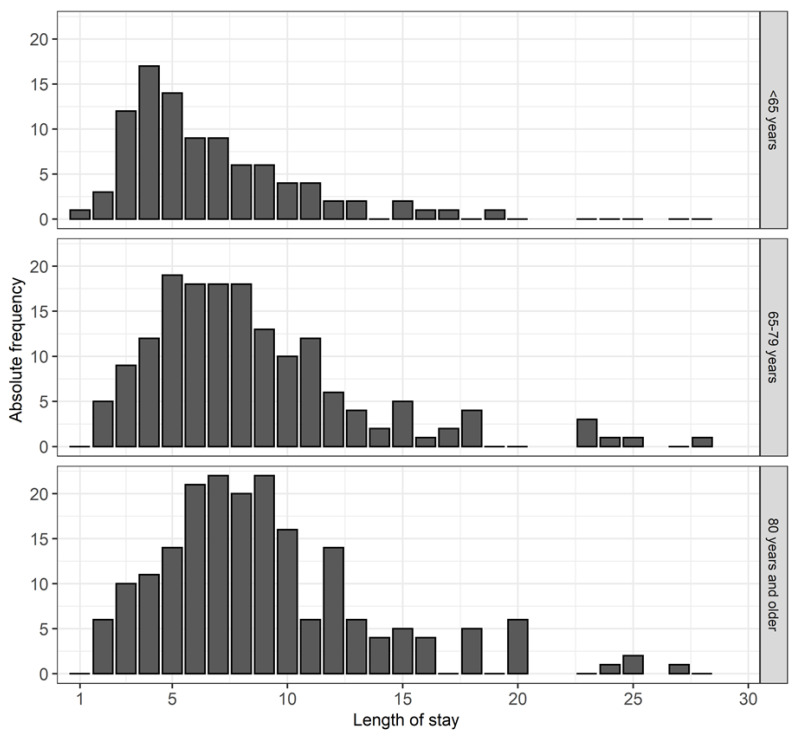
This figure shows the length of hospital stay for each age group.

**Table 1 geriatrics-07-00011-t001:** Patient characteristics for the total group and the different age categories.

Variable	Total	Age < 65 Years	Age 65–79 Years	Age ≥ 80 Years
N (%)	547 (100)	94 (21)	164 (36)	199 (44)
Age, mean (±SD)	74.0 (±15.3)	49.5 (±11.7)	73.1 (±4.1)	86.4 (±4.7)
Gender				
Male	224 (49)	36 (38)	97 (59)	91 (46)
Female	233 (51)	58 (62)	67 (41)	108 (54)
Length of stay, mean (±SD)	8.5 (±5.0)	6.6 (±3.6)	8.6 (±4.8)	9.4 (±5.6)
PCCL				
0	134 (29)	45 (48)	44 (27)	45 (23)
1	56 (12)	9 (10)	18 (11)	29 (15)
2	65 (14)	11 (12)	28 (17)	26 (13)
3	136 (30)	23 (25)	49 (30)	64 (32)
4	66 (14)	6 (6)	25 (15)	35 (18)
No. of drug interaction				
none	207 (35)	62 (66)	73 (45)	72 (36)
1–2 interactions	115 (25)	22 (23)	32 (20)	61 (31)
>2 interaction	135 (30)	10 (11)	59 (36)	66 (33)
Intensity of drug interaction				
none/mild	227 (50)	64 (70)	82 (50)	79 (40)
moderate, severe and/or contraindicated	230 (50)	28 (30)	82 (50)	120 (60)

Data are displayed as number (%) unless otherwise specified. PCCL= German Patient Clinical Complexity Level. SD = standard deviation.

**Table 2 geriatrics-07-00011-t002:** Association between hospital LOS and various factors by regression analysis.

Variable	RR	95% CI of RR	*p* Value
Intercept	4.88		
PCCL (ref: 0)			<0.001
1	1.10	0.94–1.28	
2	1.47	1.28–1–69	
3	1.36	1.20–1.53	
4	1.86	1.61–2.14	
Age (ref: <65 years)			0.017
65–79	1.09	0.97–1.24	
≥80	1.18	1.05–1.34	
No. of drug interactions	1.02	1.01–1.04	0.005
Intensity of drug interactions (ref: none/mild)			<0.001
moderate, severe, and/or contraindicated	1.22	1.09–1.37	

PCCL = German Patient Clinical Complexity Level, CI = confidence interval, SE = standard error, ref = reference.

**Table 3 geriatrics-07-00011-t003:** Association between hospital LOS and various factors for patients aged 80 years and older by regression analysis.

Variable	RR	95% CI of RR	*p* Value
Intercept	5.63		
PCCL (ref: 0)			<0.001
1	1.17	0.92–1.49	
2	1.59	1.26–2.02	
3	1.46	1.20–1.78	
4	1.73	1.39–2.16	
No. of drug interaction	1.03	1.00–1.06	0.025
Intensity of drug interactions (ref: none/mild)			0.071
moderate, severe, and/or contraindicated	1.18	0.99–1.40	

PCCL = German Patient Clinical Complexity Level, CI = confidence interval. SE = standard error, ref = reference.

## Data Availability

The datasets generated during and/or analysed during the current study are available from the corresponding author on reasonable request.

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
