# Peer review of "The Impact of Drug Interactions in Patients with Community-Acquired Pneumonia on Hospital Length of Stay"

_geriatrics, 2022, doi:10.3390/geriatrics7010011_

Round 1
Reviewer 1 Report
The paper regards the impact of drug-drug interactions on lengths of stay in people older than 65 hospitalized with community acquired pneumonia. This is a retrospective studies in one hospital, including 457 patients. This subject is of interest since the aging population will be exposed to more medications leading to more drug-drug interactions. The authors mentioned interesting tools to consider interactions and multimorbidity. The authors used the Aid-Klinik program to identify interactions, an evidence based database. Furthermore, the authors took account of the multimorbidity and the complexity of patient’s treatment when assessing the association.
The introduction presents the context and the goal of the study. The objective is stated.
Major comments
The methods paragraph does not include any reference and major elements are missing:
About the inclusion/exclusion criteria:
-Patients hospitalized with community acquired pneumonia are Identified by a uniform criteria of the IQTIG. Could you give more details? Is there a reference?
-Patients who died are excluded from the analysis. This should be mentioned in the method paragraph and the reason stated. The authors explained that this exclusion was due to a lack a comparability. This aspect should be further discussed. What are the differences? How this can have underestimated the impact of interactions that are eventually deadly?
About the variables:
The authors mentioned that 6 variables were collected but cited only four. I assumed that it also included the two variables concerning interactions when I saw table 1 but this is confusing.
The PCCL must be more described: How is it exactly calculated? How are identified the comorbidities and the complications for each patients? The list of comorbidities and complications should be included in the paper or in supplementary materials? Is there any reference?
About the Patient outcome
As the authors made a retrospective study and aimed to study the impact of drug-drug interactions on the length of stay, the outcome is not the interaction, it is the increase in length of stay. The interactions are actually the exposure.
The authors used the Aid Klinik program to identify the interactions. This must be more described. Has this tool been validated/already used in previous studies (references should be listed)? Would it be possible to include in the paper or in the supplementary materials the list of the interaction?
Which medications are considered? Are we talking about medications given prior to admission or during the inpatient stay or both? Are some medications excluded from the study?
About the results:
Results are presented for the whole cohort and people aged > 80 years old. I understand that most results were not significant for other age groups but results should be displayed, at least in supplementary materials. The fact that results were significant in the whole cohort, but not in all class aged should be addressed in the discussion.
About the discussion,
The authors listed their results in the first three paragraphs. I would expect one paragraph with the results and then the actual discussion. The results presented by the authors should be compared to previous studies about interaction in inpatients or in community patients.
Comorbidities are a major confounding factor. I would appreciate that the authors explained how this can be controlled. I understand that the analysis are adjusted for PCCL but we miss a description and a comparison of this tool with other comorbidities or frailty scores.
I do not understand the last paragraph of the discussion (225-232). Do you mean that you have also considered synergic association of drugs as interaction? Which category were they classified in? These interactions should be excluded from the model since they might have a benefit leading to an underestimation of the effect. The list of interactions considered should be available in the supplementary materials.
In the limitations paragraph (l234-238), the data of the number of drug taken by patients are missing. Does it mean that some interactions might be missing? How this could have had impacted the results?
Minor comments
226: ASS has not been defined
239-241: I do not think that it is part of the paper.
A figure in table 1 is incorrect (it is 457, not 547)
Author Response
Dear Reviewer 1,
Thank you for your constructive feedback on the article “Impact of drug interactions in patients with community-acquired pneumonia on hospital length of stay”.
Below we list the individual points that we have adapted based on your comments in the paper.
Reviewer 1 Comments:
Comments and Suggestions for Authors
The paper regards the impact of drug-drug interactions on lengths of stay in people older than 65 hospitalized with community acquired pneumonia. This is a retrospective studies in one hospital, including 457 patients. This subject is of interest since the aging population will be exposed to more medications leading to more drug-drug interactions. The authors mentioned interesting tools to consider interactions and multimorbidity. The authors used the Aid-Klinik program to identify interactions, an evidence based database. Furthermore, the authors took account of the multimorbidity and the complexity of patient’s treatment when assessing the association.
The introduction presents the context and the goal of the study. The objective is stated.
Major comments
The methods paragraph does not include any reference and major elements are missing:
Additional source references for the statistical approach, the application of the PCCL, and for evidence of the AiD Clinic Tool were added to the methodology:
References 22, 23, 24, 25, 26, 27.
About the inclusion/exclusion criteria:
-Patients hospitalized with community acquired pneumonia are Identified by a uniform criteria of the IQTIG. Could you give more details? Is there a reference?
Patients who were hospitalized with community-acquired pneumonia and were treated according to the IQTIG quality guidelines are part of this group.
We added in page 2 lines 85-87 the following sentence:
Inclusion criteria were a community-aquired pneumonia without necessary ventilation or intensive care unit stay. For a uniform and comparable treatment regiment, … [22]
-Patients who died are excluded from the analysis. This should be mentioned in the method paragraph and the reason stated. The authors explained that this exclusion was due to a lack a comparability. This aspect should be further discussed. What are the differences? How this can have underestimated the impact of interactions that are eventually deadly?
Page 2, lines 90-91; the following sentence was added:
Patients who died during the inpatient stay were removed from the analysis based on lack of comparability of the outcome (length of stay).
Based on our data situation (see limitation, limited raw data set in retrospective study design), we unfortunately cannot classify what the patients died of.
About the variables:
The authors mentioned that 6 variables were collected but cited only four. I assumed that it also included the two variables concerning interactions when I saw table 1 but this is confusing.
Page 3, lines 105-107; the following sentence was added:
In addition, the number and severity of drug interactions were collected, which are described in more detail in the following chapter.
The PCCL must be more described: How is it exactly calculated? How are identified the comorbidities and the complications for each patients? The list of comorbidities and complications should be included in the paper or in supplementary materials? Is there any reference?
Page 2-3, lines 98-101; the following sentence was added:
In the German Diagnosis Related Group (G-DRG) classification, complications and/or comorbidities (CC) are mapped using the patient-related total severity code (PCCL). The PCCL is calculated from the cumulative severities of complications and/or comorbidities (CCL) of a patient's individual.
Page 3, lines 101-102; the following sentence was deleted:
The PCCL is calculated from the severity of the comorbidities and complications of the respective patient.
The Institute for Hospital Remuneration Systems (InEK) publishes a definition manual every year in which the current calculation bases are explained. The procedure for PCCL determination is always identical. Unfortunately, we can only refer to the German website of the InEK (InEK Website has been added as reference (page 3, line 99).
About the Patient outcome
As the authors made a retrospective study and aimed to study the impact of drug-drug interactions on the length of stay, the outcome is not the interaction, it is the increase in length of stay. The interactions are actually the exposure.
We deleted the first sentence under chapter 2.3 “Patient outcome”. It is correct that LOS is the primary endpoint.
The authors used the Aid Klinik program to identify the interactions. This must be more described. Has this tool been validated/already used in previous studies (references should be listed)?
The AiD Clinic® tool is provided commercially to hospitals and can be used as a stand alone drug information system or can be used as patient related alerts when integrated with the hospital system. There are several studies that also refer to the AiD Klinik® Tool from the University in Heidelberg. Unfortunately, there is no validation study that we can cite in this regard, but here are several references of studies using the AiD Klinik® Tool.
- Impact of pharmacist interventions in older patients: a prospective study in a tertiary hospital in Germany; doi: 10.2147/CIA.S109048
- Potential Drug-Drug Interactions in a Cohort of Elderly, Polymedicated Primary Care Patients on Antithrombotic Treatment; doi: 10.1007/s40266-018-0550-6
- The expenditure of computer-related worktime using clinical decision support systems in chronic pain therapy; doi: 10.1186/s12871-015-0094-9
A reference about the AiD Klinik® Tool has been added. (page 3 line 112, reference number 24, 25)
Would it be possible to include in the paper or in the supplementary materials the list of the interaction?
Unfortunately not, because we did not make a list of the interactions. It has been noticed how many interactions occurred per patient during treatment and how many mild/moderate/severe/contraindicated interactions per patient, but not which concrete combinations of medication.
Which medications are considered? Are we talking about medications given prior to admission or during the inpatient stay or both? Are some medications excluded from the study?
As described in lines 109 to 110, all medications given during the inpatient stay were examined.
About the results:
Results are presented for the whole cohort and people aged > 80 years old. I understand that most results were not significant for other age groups but results should be displayed, at least in supplementary materials. The fact that results were significant in the whole cohort, but not in all class aged should be addressed in the discussion.
Based on your recommendation, we added the regression results for the two additional age groups (<65 years and 65-79 years) to the supplements (Page 10-11).
About the discussion,
The authors listed their results in the first three paragraphs. I would expect one paragraph with the results and then the actual discussion. The results presented by the authors should be compared to previous studies about interaction in in-patients or in community patients.
Page 7, lines 207-209 added:
The higher number of secondary diagnoses explains the fact that the older generation is more frequently affected by drug interactions. This is also confirmed by the existing literature [29,30] .
Page 7, lines 218-221 we added the following:
… and the results vary between 0.25 and 7 days [31,32] . However, it should be noted that the published sources have different patient cohorts (internal medicine and surgery). However, these studies, as well as our results, suggest that drug interactions have an impact on the length of hospital stay of patients.
Comorbidities are a major confounding factor. I would appreciate that the authors explained how this can be controlled. I understand that the analysis are adjusted for PCCL but we miss a description and a comparison of this tool with other comorbidities or frailty scores.
The PCCL is a score based on the German billing system. Since this is generated by a complex method through the billing system, there is unfortunately no comparability to a clearly defined frailty score.
On page 2 to 3 we added (as described in the comments before) a more detailed description.
I do not understand the last paragraph of the discussion (225-232). Do you mean that you have also considered synergic association of drugs as interaction? Which category were they classified in? These interactions should be excluded from the model since they might have a benefit leading to an underestimation of the effect. The list of interactions considered should be available in the supplementary materials.
We checked all medications a patient received during treatment for interactions. The names of medications that interacted were not listed. Moreover, we did not differ between synergic associations, because for example a synergic, intended drug interaction such as Acetylsalicylsäure (ASS) / Rivaroxaban on the other hand leads especially in combination with a further interaction to an increased risk for bleeding. However, it would be of future interest to differentiate the interactions in avoidable und not avoidable interactions to figure out if the part of avoidable drug interactions has even a greater impact on the LOS.
In the limitations paragraph (l234-238), the data of the number of drug taken by patients are missing. Does it mean that some interactions might be missing? How this could have had impacted the results?
This seems to have been written by us in a misleading way. There is no influence on the results. We have adapted the sentences in the Limitations as follows:
Page 7, lines 251-252; the following sentence was changed into:
This retrospective data analysis has limitations that need to be mentioned here. On the one hand, the number of drugs taken per patient is missing from the present raw data set.
Page 7, lines 252-254; the following sentence was adapted:
However, when using the AiD tool to generate the number and severity of drug interactions, these were completely available in the clinic.
Minor comments
226: ASS has not been defined
Acetylsalicylsäure (ASS) was adapted
239-241: I do not think that it is part of the paper.
Page 8, lines 257-259 were deleted.
A figure in table 1 is incorrect (it is 457, not 547)
Figure in table 1 was corrected.
Reviewer 2 Report
The study has investigated the influence of drug interactions on the length of stay in hospital. In my opinion, despite this current evaluation of the manuscript, the information reported needs complete clarification. Overall, the manuscript seems vague and imprecise in many sections (e.g., some sentences are redundant). I have a number of recommendations that I think may improve the manuscript.
- The title and the aim of the study do not match. The authors say incidence and severity in the title, but the aims are towards influence/factors. Please clarify
- What’s the rationale behind choosing patients with community-acquired pneumonia for studying drug interactions?
- These days there are several measures to prevent/avoid drug interactions. Why do the authors believe that drug interactions are prevalent in hospitalised patients and how drug interactions can impact the LOS (according to the literature)? I agree with the fact that multimorbidity
- Why do the authors not consider LOS as a linear variable, and would risk ratio be more relevant in this type of study? E.g., Multivariate linear regression, adjusted by sex, age group and clinical condition, could be employed to assess whether the length of stay was influenced by the severity of drug interaction.
- I agree with the statement made by authors – “Individuals who are older than 65 years or have a chronic health condition are considered to be at high risk for pneumonia”. However, the only exclusion criterion was age below 18 years. Most of your discussions are around older adults. Please clarify.
- We don’t normally present the data with parameter estimates and SE. Table 3 is useful to be supplementary data.
- Why there is no data on what medications (and the number of meds) contributed to drug interactions?
Author Response
Dear Reviewer 2,
Thank you for your constructive feedback on the article “Impact of drug interactions in patients with community-acquired pneumonia on hospital length of stay”.
Below we list the individual points that we have adapted based on your comments in the paper.
Comments and Suggestions for Authors
The study has investigated the influence of drug interactions on the length of stay in hospital. In my opinion, despite this current evaluation of the manuscript, the information reported needs complete clarification. Overall, the manuscript seems vague and imprecise in many sections (e.g., some sentences are redundant). I have a number of recommendations that I think may improve the manuscript.
- The title and the aim of the study do not match. The authors say incidence and severity in the title, but the aims are towards influence/factors. Please clarify
We adapted the title as follows:
“Impact of drug interactions in patients with community-acquired pneumonia on hospital length of stay”.
What’s the rationale behind choosing patients with community-acquired pneumonia for studying drug interactions?
Patients with pneumonia that need an in-hospital treatment tend to be multimorbide and receive more various medications, than for example patients with gastritis (Reference Introduction number 18 and 19). Moreover, a pneumonia is a disease with a high prevalence.
Furthermore, in one study excessive polypharmacy was identified as a risk factor for pneumonia (Factors associated with hospitalization for community-acquired
pneumonia in home health care patients in Taiwan, https://doi.org/10.1007/s40520-019-01169-8).
These days there are several measures to prevent/avoid drug interactions. Why do the authors believe that drug interactions are prevalent in hospitalised patients and how drug interactions can impact the LOS (according to the literature)? I agree with the fact that multimorbidity.
The literature repeatedly confirms that adverse drug reactions play a major role, especially in hospitals (Adverse Drug Reactions in an Elderly Hospitalised Population, https://doi.org/10.2165/00002512-200522090-00005). Based on multimorbidity in combination with an increased age, the number of necessary drugs increases. Furthermore, there is the statement that a hospital stay, or the length of a hospital stay, carries further risks for the patients (complications such as reduced mobility, infections, and so on) (National Health Services Improvement. Guide to reducing long hospital stays, https://improvement.nhs.uk/documents/2898/Guide_to_reducing_long_hospital_stays_FINAL_v2.pdf). In addition, the recurrently discussed resource limitations of hospitals are an issue. Thus, for us authors, considering the impact of drug interaction in the hospital using the example of community-acquired pneumonia is an important topic to exemplify the complexity.
Why do the authors not consider LOS as a linear variable, and would risk ratio be more relevant in this type of study? E.g., Multivariate linear regression, adjusted by sex, age group and clinical condition, could be employed to assess whether the length of stay was influenced by the severity of drug interaction.
Thank you. We did not use multiple linear regression analysis as this type of regression analysis assumes a continuous variable as response. Yet, length of stay (LOS) is a non-negative integer variable (=count data) and tends to be right-skewed in our sample (see also Figure 1). In general, count data are characterized by a type of data in which the observation represents the number of certain events and LOS can be seen as one example of this (->Hilbe-Reference). Models based on discrete distributions that take special properties of count data into account are more appropriate.
Page 3, line 130 the following reference [27] was added:
Hilbe J.M. (2011) Modeling Count Data. In: Lovric M. (eds) International Encyclopedia of Statistical Science. Springer, Berlin, Heidelberg. https://doi.org/10.1007/978-3-642-04898-2_369
I agree with the statement made by authors – “Individuals who are older than 65 years or have a chronic health condition are considered to be at high risk for pneumonia”. However, the only exclusion criterion was age below 18 years. Most of your discussions are around older adults. Please clarify.
We wanted to notice the distribution pattern of patients who were treated and we intended to compare geriatric patients to the others.
That most of the patients were older adults is due to fact that older adults are more likely to be multimorbide und need more likely a hospital treatment than younger ones.
We don’t normally present the data with parameter estimates and SE. Table 3 is useful to be supplementary data.
We have removed "beta" and "SE" from Tables 2 and 3.
Based on the significant results, we would like to leave Table 3 in the paper. However, we still added the regression results of the patient groups <65 years and 65-7 years in the supplements.
Why there is no data on what medications (and the number of meds) contributed to drug interactions?
As listed in the limitations, the number of drugs as well as their drug names are not part of the present data set in the retrospective study design. Only the number of interactions per patient during treatment and the number of mild/medium/severe/contraindicated interactions per patient were determined, but not the specific drug combinations.
Round 2
Reviewer 1 Report
2nd review: The authors properly addressed most comments. However some elements still need to be completed or further discussed in the manuscript. I answer “ok” to properly addressed comments and made additional comments in this review.
Dear Reviewer 1,
Thank you for your constructive feedback on the article “Impact of drug interactions in patients with community-acquired pneumonia on hospital length of stay”.
Below we list the individual points that we have adapted based on your comments in the paper.
Reviewer 1 Comments:
Comments and Suggestions for Authors
The paper regards the impact of drug-drug interactions on lengths of stay in people older than 65 hospitalized with community acquired pneumonia. This is a retrospective studies in one hospital, including 457 patients. This subject is of interest since the aging population will be exposed to more medications leading to more drug-drug interactions. The authors mentioned interesting tools to consider interactions and multimorbidity. The authors used the Aid-Klinik program to identify interactions, an evidence based database. Furthermore, the authors took account of the multimorbidity and the complexity of patient’s treatment when assessing the association.
The introduction presents the context and the goal of the study. The objective is stated.
Major comments
The methods paragraph does not include any reference and major elements are missing:
Additional source references for the statistical approach, the application of the PCCL, and for evidence of the AiD Clinic Tool were added to the methodology:
References 22, 23, 24, 25, 26, 27.
About the inclusion/exclusion criteria:
-Patients hospitalized with community acquired pneumonia are Identified by a uniform criteria of the IQTIG. Could you give more details? Is there a reference?
Patients who were hospitalized with community-acquired pneumonia and were treated according to the IQTIG quality guidelines are part of this group.
We added in page 2 lines 85-87 the following sentence:
Inclusion criteria were a community-aquired pneumonia without necessary ventilation or intensive care unit stay. For a uniform and comparable treatment regiment, … [22]
Answer 2nd review: Ok
-Patients who died are excluded from the analysis. This should be mentioned in the method paragraph and the reason stated. The authors explained that this exclusion was due to a lack a comparability. This aspect should be further discussed. What are the differences? How this can have underestimated the impact of interactions that are eventually deadly?
Page 2, lines 90-91; the following sentence was added:
Patients who died during the inpatient stay were removed from the analysis based on lack of comparability of the outcome (length of stay).
Based on our data situation (see limitation, limited raw data set in retrospective study design), we unfortunately cannot classify what the patients died of.
Answer 2nd review: Ok
About the variables:
The authors mentioned that 6 variables were collected but cited only four. I assumed that it also included the two variables concerning interactions when I saw table 1 but this is confusing.
Page 3, lines 105-107; the following sentence was added:
In addition, the number and severity of drug interactions were collected, which are described in more detail in the following chapter.
Answer 2nd review: Ok
The PCCL must be more described: How is it exactly calculated? How are identified the comorbidities and the complications for each patients? The list of comorbidities and complications should be included in the paper or in supplementary materials? Is there any reference?
Page 2-3, lines 98-101; the following sentence was added:
In the German Diagnosis Related Group (G-DRG) classification, complications and/or comorbidities (CC) are mapped using the patient-related total severity code (PCCL). The PCCL is calculated from the cumulative severities of complications and/or comorbidities (CCL) of a patient's individual.
Page 3, lines 101-102; the following sentence was deleted:
The PCCL is calculated from the severity of the comorbidities and complications of the respective patient.
The Institute for Hospital Remuneration Systems (InEK) publishes a definition manual every year in which the current calculation bases are explained. The procedure for PCCL determination is always identical. Unfortunately, we can only refer to the German website of the InEK (InEK Website has been added as reference (page 3, line 99).
Answer 2nd review: Ok
About the Patient outcome
As the authors made a retrospective study and aimed to study the impact of drug-drug interactions on the length of stay, the outcome is not the interaction, it is the increase in length of stay. The interactions are actually the exposure.
We deleted the first sentence under chapter 2.3 “Patient outcome”. It is correct that LOS is the primary endpoint.
Answer 2nd review: Ok
The authors used the Aid Klinik program to identify the interactions. This must be more described. Has this tool been validated/already used in previous studies (references should be listed)?
The AiD Clinic® tool is provided commercially to hospitals and can be used as a stand alone drug information system or can be used as patient related alerts when integrated with the hospital system. There are several studies that also refer to the AiD Klinik® Tool from the University in Heidelberg. Unfortunately, there is no validation study that we can cite in this regard, but here are several references of studies using the AiD Klinik® Tool.
- Impact of pharmacist interventions in older patients: a prospective study in a tertiary hospital in Germany; doi: 10.2147/CIA.S109048
- Potential Drug-Drug Interactions in a Cohort of Elderly, Polymedicated Primary Care Patients on Antithrombotic Treatment; doi: 10.1007/s40266-018-0550-6
- The expenditure of computer-related worktime using clinical decision support systems in chronic pain therapy; doi: 10.1186/s12871-015-0094-9
A reference about the AiD Klinik® Tool has been added. (page 3 line 112, reference number 24, 25)
Answer 2nd review: This should be added in the discussion/limitations
Would it be possible to include in the paper or in the supplementary materials the list of the interaction?
Unfortunately not, because we did not make a list of the interactions. It has been noticed how many interactions occurred per patient during treatment and how many mild/moderate/severe/contraindicated interactions per patient, but not which concrete combinations of medication.
Answer 2nd review: This should be added in the discussion/limitations
Which medications are considered? Are we talking about medications given prior to admission or during the inpatient stay or both? Are some medications excluded from the study?
As described in lines 109 to 110, all medications given during the inpatient stay were examined.
Answer 2nd review: Ok
About the results:
Results are presented for the whole cohort and people aged > 80 years old. I understand that most results were not significant for other age groups but results should be displayed, at least in supplementary materials. The fact that results were significant in the whole cohort, but not in all class aged should be addressed in the discussion.
Based on your recommendation, we added the regression results for the two additional age groups (<65 years and 65-79 years) to the supplements (Page 10-11).
Answer 2nd review: L 189 “data not shown” should be removed. The fact that results were significant in the whole cohort but not in every age class should be addressed in the discussion. This should also be stated in the conclusion.
About the discussion,
The authors listed their results in the first three paragraphs. I would expect one paragraph with the results and then the actual discussion. The results presented by the authors should be compared to previous studies about interaction in in-patients or in community patients.
Page 7, lines 207-209 added:
The higher number of secondary diagnoses explains the fact that the older generation is more frequently affected by drug interactions. This is also confirmed by the existing literature [29,30] .
Page 7, lines 218-221 we added the following:
… and the results vary between 0.25 and 7 days [31,32] . However, it should be noted that the published sources have different patient cohorts (internal medicine and surgery). However, these studies, as well as our results, suggest that drug interactions have an impact on the length of hospital stay of patients.
Answer 2nd review: L 234 -235, add the references to “These findings are also supported by existing literature.”
Comorbidities are a major confounding factor. I would appreciate that the authors explained how this can be controlled. I understand that the analysis are adjusted for PCCL but we miss a description and a comparison of this tool with other comorbidities or frailty scores.
The PCCL is a score based on the German billing system. Since this is generated by a complex method through the billing system, there is unfortunately no comparability to a clearly defined frailty score.
On page 2 to 3 we added (as described in the comments before) a more detailed description.
Answer 2nd review: This should be added to the discussion/limitations.
I do not understand the last paragraph of the discussion (225-232). Do you mean that you have also considered synergic association of drugs as interaction? Which category were they classified in? These interactions should be excluded from the model since they might have a benefit leading to an underestimation of the effect. The list of interactions considered should be available in the supplementary materials.
We checked all medications a patient received during treatment for interactions. The names of medications that interacted were not listed. Moreover, we did not differ between synergic associations, because for example a synergic, intended drug interaction such as Acetylsalicylsäure (ASS) / Rivaroxaban on the other hand leads especially in combination with a further interaction to an increased risk for bleeding. However, it would be of future interest to differentiate the interactions in avoidable und not avoidable interactions to figure out if the part of avoidable drug interactions has even a greater impact on the LOS.
Answer 2nd review: Thank you for clarifying. I suggest that the authors complete the discussion with elements from this answer. The example helps to understand.
In the limitations paragraph (l234-238), the data of the number of drug taken by patients are missing. Does it mean that some interactions might be missing? How this could have had impacted the results?
This seems to have been written by us in a misleading way. There is no influence on the results. We have adapted the sentences in the Limitations as follows:
Page 7, lines 251-252; the following sentence was changed into:
This retrospective data analysis has limitations that need to be mentioned here. On the one hand, the number of drugs taken per patient is missing from the present raw data set.
Page 7, lines 252-254; the following sentence was adapted:
However, when using the AiD tool to generate the number and severity of drug interactions, these were completely available in the clinic.
Answer 2nd review: Ok
Minor comments
226: ASS has not been defined
Acetylsalicylsäure (ASS) was adapted
Answer 2nd review: The English words “Aspirin” or “acetylsalicylic acid” should be preferred in this context. The authors should remove ASS if this abbreviation is not used later.
239-241: I do not think that it is part of the paper.
Page 8, lines 257-259 were deleted.
Answer 2nd review: Ok
A figure in table 1 is incorrect (it is 457, not 547)
Figure in table 1 was corrected
Answer 2nd review: Ok
Author Response
Dear Reviewer,
Thank you for the detailed assistance on the outstanding points of the reviewed article. We have worked on the individual points as follows:
- The authors used the Aid Klinik program to identify the interactions. This must be more described. Has this tool been validated/already used in previous studies (references should be listed)?
- Answer 2nd review: This should be added in the discussion/limitations
We added the following statement to the limitations part of the manuscript (line 265-266):
Although the AiD tool has been used in several clinical trials, a validation study for the tool itself has not been published.
- Would it be possible to include in the paper or in the supplementary materials the list of the interaction?
- Answer 2nd review: This should be added in the discussion/limitations
We added to the limitations in the sentence stating that the number of drugs taken per patient is not documented, that there is no list of all interactions. (line 259-260):
as well as a list of interactions
- Answer 2nd review: L 189 “data not shown” should be removed. The fact that results were significant in the whole cohort but not in every age class should be addressed in the discussion. This should also be stated in the conclusion.
We removed "data not shown" in line 189
As the results were significant for the whole cohort, we think that the conclusion can be drawn in the same way as it is.
- Answer 2nd review: L 234 -235, add the references to “These findings are also supported by existing literature.”
We added references 31 and 32.
- Comorbidities are a major confounding factor. I would appreciate that the authors explained how this can be controlled. I understand that the analysis are adjusted for PCCL but we miss a description and a comparison of this tool with other comorbidities or frailty scores.
- Author: The PCCL is a score based on the German billing system. Since this is generated by a complex method through the billing system, there is unfortunately no comparability to a clearly defined frailty score. On page 2 to 3 we added (as described in the comments before) a more detailed description.
- Answer 2nd review: This should be added to the discussion/limitations.
We added "using the PCCL score" into the following sentence of the limitations part:
Furthermore, the known variables of multimorbidity using the PCCL score and age were included in the statistical analyses, but there is the possibility that other confounders not recorded here influence the outcome of length of stay.
- I do not understand the last paragraph of the discussion (225-232). Do you mean that you have also considered synergic association of drugs as interaction? Which category were they classified in? These interactions should be excluded from the model since they might have a benefit leading to an underestimation of the effect. The list of interactions considered should be available in the supplementary materials.
- Answer 2nd review: Thank you for clarifying. I suggest that the authors complete the discussion with elements from this answer. The example helps to understand.
In the discussion the examples for avoidable and intended drug interactions is given as follows:
In this retrospective study, no distinction was made between drug interactions that were intentionally prescribed, such as acetylsalicylic acid and Rivaroxaban in the treatment of coronary heart disease, and avoidable drug interactions, such as atorvastatin and clarithromycin. In this regard, studies should follow to differentiate between avoidable and unavoidable drug interactions. (Lines 249-253)
- Answer 2nd review: The English words “Aspirin” or “acetylsalicylic acid” should be preferred in this context. The authors should remove ASS if this abbreviation is not used later.
ASS was removed (line 250)
Reviewer 2 Report
Thanks for carefully addressing the comments.
Author Response
Dear Reviewer,
thank you for reviewing my article. As I can see, there is no open question anymore. I hope I didn´t oversee anything?
Best regards and merry christmas!